# Morphology and Characterisation of Novolac–LDPE-Based Mixtures as Matrix for Injection Moulded Green Bodies for Bio-Based SiC Ceramics †

**Matthias Mihalic [1,]***, **Fernanda Soares de Sousa [1]**, **Ivana Burzic [1]**, **Andreas Hinterreiter [2]**, **David Stifter [2] and Christian Fürst [1]**

[1]   Wood K plus-Kompetenzzentrum Holz GmbH, Division Biobased Composites and Processes, Altenberger Strasse 69, 4040 Linz, Austria
[2]   Center for Surface and Nanoanalytics (ZONA), Johannes Kepler University Linz, Altenberger Strasse 69, 4040 Linz, Austria
*   Correspondence: m.mihalic@wood-kplus.at
†   This paper is an expanded version of a conference paper which has been presented at: Mihalic, M.; Soares de Sousa, F.; Burzic, I.; Hinterberger, A.; Stifter, D.; Fürst, C. Morphology and Characterisation of Novolac-LDPE Based Mixtures as Matrix for Injection Moulded Green Bodies for Bio-based SiC Ceramics. In Proceedings of the ICCST/12—12th International Conference of Composite Science and Technology, Sorrento, Italy, 8–10 May 2019.

**Abstract:** This work focuses on the influence of the composition of novolac–LDPE-based mixtures, which serve as a matrix for the green bodies for bio-based silicon carbide (C/Si/SiC) ceramics, on the morphology and the mechanical properties of the green bodies and the ceramics produced thereof. The green bodies were obtained through compounding and injection moulding, and were characterised by scanning electron microscopy (SEM) and mechanical testing. Selected formulations were reinforced with natural fibres, pyrolysed to yield porous carbon templates, and converted into C/Si/SiC ceramics via liquid silicon infiltration. The carbon and ceramic specimens were characterised by light optical microscopy (LOM) and mechanical testing. Without further additives, very coarse morphologies of the novolac–LDPE-based mixtures were obtained, but the miscibility could be improved by the addition of a coupling agent and a lubricant. The pore structure of the carbon specimens was dependent on the phase distribution in the green bodies, and in turn determined the morphology of the C/Si/SiC ceramics. In all steps of the process chain, the morphology had a very strong influence on the mechanical properties. From green bodies with a homogeneous phase distribution, ceramic specimens with a SiC content of up to 75 vol% could be obtained.

**Keywords:** silicon carbide; natural fibre composites; injection moulding; bio-based; morphology; mechanical properties

---

## 1. Introduction

Silicon carbide (SiC) is one of the most versatile engineering ceramics because of its high hardness, chemical resistance, and heat resistance. However, its potential for broader use is limited by its very high cost of materials and processing. A promising approach towards cost reduction, and thus towards widening the scope of potential applications, is the development of bio-based C/Si/SiC ceramics. These can be produced in a three-step process: (1) shaping green bodies from natural fibre composites; (2) pyrolysis of the green bodies into porous C-templates; (3) liquid silicon infiltration of the C-templates to yield C/Si/SiC ceramics.

Two main routes for the production of the green bodies have been reported: Either from thermoset bonded natural fibre boards (chipboards, MDF) [1–4], or from thermoplastic-based wood polymer composites (WPC) by extrusion or injection moulding [5]. The first method is mainly limited to planar geometries and requires costly and time-consuming further processing to obtain three-dimensional (3D) structures. The second method makes 3D geometries easily accessible, but the common thermoplastic matrices of WPC offer low carbon yield and limited dimensional stability during carbonisation.

A third route which eliminates the drawbacks of the aforementioned methods has been developed at Wood K plus: by extrusion of phenolic resin-based WPC into green bodies, high carbon yield and dimensional stability in the carbonisation process are achieved. The porous structure of the carbon specimens, which is necessary for the liquid silicon infiltration, is obtained by adding thermoplastic modifiers as placeholders in the green body, which decompose almost completely upon carbonisation [6].

The homogeneity of the composites forming the green bodies is of paramount importance for the distribution of carbon and pores in the porous C-templates, which in turn strongly influences the silicon infiltration and conversion into SiC. Therefore, an optimal compounding process prior to the green body formation is required [6,7].

The main constituents of the green bodies play specific roles in the process [8]. The thermoset component, in combination with a hardening agent, is mainly responsible for the formation of carbon during the pyrolysis and, in a cured state, offers structural stability. The thermoplastic component facilitates processing and is responsible for the pore formation upon pyrolysis. The fibres (wood or cellulose) offer structural stability, especially in the early stages of the carbonisation, and later yield additional carbon. Important processing additives are coupling agents (for an improved compatibility and adhesion between the fibres and the polymeric phases) and lubricants (for ease of processing especially during the formation of the green bodies). The total amount and distribution of thermoplastic material in the composite determine the bulk density, respectively the porosity, of the porous carbon specimens. This in turn affects the amount of silicon which reacts with the carbon during the silicon infiltration process; depending on the size and distribution of the pores, smaller or larger areas of pure carbon and/or pure silicon may still be present in the obtained SiC specimens.

The aim of this work was to study the phase distribution in the green bodies depending on the composition, and its effect on the phase distribution and properties of the carbon specimens and C/Si/SiC ceramics. The influence of the ratio of the matrix components on the morphology and mechanical properties was first investigated in compounds without fibres. Based on these observations, selected formulations containing fibres were prepared for the subsequent pyrolysis and silicon infiltration. A comparison of the different formulations throughout the experimental chain (green bodies without fibres—green bodies with fibres—porous carbon specimens—ceramic specimens) clearly shows the influence of the composition on the morphology and the mechanical properties.

## 2. Materials and Methods

### 2.1. Materials

The basic formulation for all materials in this study consisted of a novolac resin (Prefere 824442X from Prefere Resins) and a low density polyethylene (LDPE—CA8200 from Borealis Polyolefine GmbH) in varying ratios. Hexamethylenetetramine (HMTA) from Ineos Paraform was added as hardener; its content was kept constant at 4 m% respective to the (novolac + HMTA) content in the mixture. The decision to use LDPE as the thermoplastic component was made mainly due to its relatively low melting temperature of 108 °C, with the aim of keeping the processing temperatures of the compounds low enough to limit the degree of hardening of the (novolac + HMTA) phase during processing.

Selected formulations contained Mowital B30T (polyvinylbutyral) from Kuraray Europe GmbH as compatibiliser/coupling agent, and/or Naftosafe PHX 369D from Chemson Polymer-Additive AG as lubricant. Where applicable, a 1:1 mixture of bleached Kraft pulp cellulose fibres and wood flour was added as filler.

The experimental programme was organised into three consecutive series. A detailed description of the formulations prepared for each series is given in the respective sections of this paper:

- Series 1—Influence of different (novolac + HMTA)–LDPE ratios on the compound properties, without further additives (cf. Section 3.1.).
- Series 2—Influence of the addition of coupling agent and lubricant (cf. Section 3.2).
- Series 3—Influence of the addition of fibres (cf. Section 3.3.).

*2.2. Processing*

2.2.1. Preparation of the Green Bodies

The compounds were prepared on a Brabender DSE20/40D twin screw extruder with a 20 mm screw diameter. A throughput of 4 kg/h and a screw speed of 200 rpm were chosen for all samples. Preliminary tests revealed a residence time of approximately 40 s under these conditions. Barrel temperatures of 145 °C or 135 °C were chosen depending on the composition—see Sections 3.1–3.3, respectively. The melt temperature, melt pressure, and torque during compounding were recorded.

For the mechanical and microscopic characterisation and the subsequent carbonisation and silicon infiltration, the compounds were injection moulded into standard shoulder bars according to ISO 527-2 on a Battenfeld HM1300/350 injection moulding machine. A barrel temperature rising from 130 °C at the hopper to 140 °C just before the nozzle, a nozzle temperature of 150 °C, and a mould temperature of 40 °C were chosen. The maximum injection pressure was recorded. The residence time during injection moulding was estimated to approximately 140 s, and thus much longer than the residence time during compounding. The recorded processing parameters for each formulation are listed in the respective sections.

2.2.2. Carbonisation and Liquid Silicon Infiltration

The green bodies from Series 3 were carbonised in a Gero HTK-8 high temperature furnace at 900 °C under nitrogen atmosphere, in order to obtain porous carbon specimens. Subsequently, silicon carbide (SiC) specimens were obtained through silicon infiltration at 1600 °C under vacuum, again in the Gero HTK-8 furnace.

*2.3. Material Characterisation*

2.3.1. Characterisation of the Green Bodies

A three-point bending test according to ISO 178 was carried out on a Messphysik BETA5 universal testing machine at a test speed of 3 mm/min. A Charpy unnotched impact test according to DIN EN ISO 179-1eU was performed on an Instron CEAST 9050 impact pendulum with a pendulum energy of 0.5 J. Six specimens of each formulation were used for the bending test, and ten for the impact test.

The morphology of the green bodies was investigated by scanning electron microscopy (SEM) of the cross section of the injection moulded specimens, in combination with energy dispersive X-ray spectroscopy (EDX) to identify the phases, using a Phenom ProX desktop microscope. A smooth sample surface for the SEM was prepared using a Leica RM2155 microtome. The EDX analysis was supplemented by Raman spectroscopy, using a Horiba LabRAM Aramis confocal Raman microscope.

2.3.2. Characterisation of the Carbon and Ceramic Specimens

The carbon and SiC specimens were characterised by a three-point bending test according to ISO 178 (using three specimens for each test), optical microscopy of the polished cross sections, and density measurements. The density of the SiC specimens was determined by applying Archimedes' principle according to DIN EN ISO 1183-1 using a Sartorius BP 301 s Analytical Balance with a YDK 01 Density Kit. This was not possible for the carbon specimens due to their porosity; for these, the bulk density was calculated via the mass and volume of rectangular shaped pieces cut from the test bars.

## 3. Results and Discussion

### 3.1. Series 1—Influence of Different (Novolac + HMTA)–LDPE Ratios

The objective of the first stage of this study was the investigation of the influence of the novolac–LDPE ratio on the compound properties. For that purpose, five formulations were prepared with the mass ratio of (novolac + HMTA) to LDPE ranging from 80:20 to 30:70. The exact compositions are listed in Table 1. Note that all contents are listed in mass percentages (m%) as well as in volume percentages (vol%). The volume percentage of a component is actually a more significant indicator of its influence on the morphology and mechanical properties of a composite. However, the compositions in this study are defined in mass percentages according to industrial standards and due to practical reasons, most importantly the gravimetric dosage during compounding.

**Table 1.** List of the formulations prepared for Series 1.

| Sample Code | Mass Ratio (Novolac + HMTA)–LDPE | | Novolac | HMTA | LDPE |
|:---:|:---:|:---:|:---:|:---:|:---:|
| S1-1 | 80:20 | (m%) | 76.8 | 3.2 | 20.0 |
| | | (vol%) | 71.8 | 2.8 | 25.4 |
| S1-2 | 70:30 | (m%) | 67.2 | 2.8 | 30.0 |
| | | (vol%) | 60.8 | 2.4 | 36.9 |
| S1-3 | 60:40 | (m%) | 57.6 | 2.4 | 40.0 |
| | | (vol%) | 50.4 | 2.0 | 47.6 |
| S1-4 | 50:50 | (m%) | 48.0 | 2.0 | 50.0 |
| | | (vol%) | 40.7 | 1.6 | 57.7 |
| S1-5 | 30:70 | (m%) | 28.8 | 1.2 | 70.0 |
| | | (vol%) | 23.0 | 0.9 | 76.1 |

In the previous studies on extruded fibre-reinforced novolac–LDPE-based green bodies that led to the present work, a barrel temperature of 135 °C was used; however, preliminary experiments for this study showed that by increasing the barrel temperature to 145 °C, a much better processability of the compounds was achieved as long as no fibres were added. Hence, a barrel temperature of 145 °C was used for all formulations of Series 1 and Series 2. The recorded values of torque, melt temperature, and melt pressure during compounding, the maximum injection pressure during injection moulding, as well as the results from the mechanical characterisation, are listed in Table 2.

Generally, the poor compatibility between the LDPE and the novolac-based phase led to fluctuations in the processing parameters, which is most obvious in the melt pressure recorded for the formulation S1-2. During compounding, the melt temperature (recorded at the end of the barrel) significantly exceeded the barrel temperature, thereby promoting the hardening process of the novolac-based phase, the effects of which are discussed along with the morphology in Section 3.1.1.

**Table 2.** Recorded processing parameters for the compounds, and mechanical properties of the green bodies of Series 1.

| Sample Code | *Compounding (145 °C)* | | | *Injection Moulding* | *Mechanical Properties* | | |
|:---:|:---:|:---:|:---:|:---:|:---:|:---:|:---:|
| | Torque | Melt Temp | Melt Pressure | Max. Injection Pressure | Flexural Modulus | Flexural Strength | Impact Strength |
| | (Nm) | (°C) | (bar) | (bar) | (GPa) | (MPa) | (kJ/m$^2$) |
| S1-1 | 18 | 156 | 18 | 490–630 | 3.37 ± 0.06 | 8.8 ± 1.1 | 0.9 ± 0.1 |
| S1-2 | 18 | 158 | 7–30 | 970–1000 | 1.99 ± 0.11 | 8.8 ± 0.2 | 2.5 ± 0.4 |
| S1-3 | 15 | 159 | 11 | 880 | 1.54 ± 0.15 | 11.3 ± 0.8 | 5.3 ± 0.7 |
| S1-4 | 15 | 159 | 21 | 860–960 | 0.80 ± 0.11 | 10.3 ± 0.7 | 17.9 ± 2.9 |
| S1-5 | 19 | 157 | 12 | 790 | 0.55 ± 0.00 | 10.8 ± 0.6 | 83.1 ± 2.6 |

### 3.1.1. Morphology

Figure 1 shows the SEM images of the cross-sections of the injection moulded specimens. While at 20 m% LDPE the morphology is made up of a novolac-based matrix with dispersed LDPE particles (according to the EDX analysis, see below), a phase inversion appears to occur at 30 m% LDPE content; at even higher LDPE contents, LDPE clearly forms the continuous phase. Generally, the morphologies of the samples appear very coarse, especially for (novolac + HMTA) to LDPE ratios of 60:40 and 50:50, where chunks of the novolac-based phase of several hundred microns in size were found. This is a clear indication that the (novolac + HMTA) phase underwent a significant degree of hardening during processing, and therefore was no longer present in the molten state but instead acted as a solid filler for the LDPE phase. The opposite is true for the formulation S1-1, which allowed the formation of a continuous novolac-based phase. This is also reflected in the strong increase of the injection pressure between the formulations S1-1 and S1-2 (cf. Table 2).

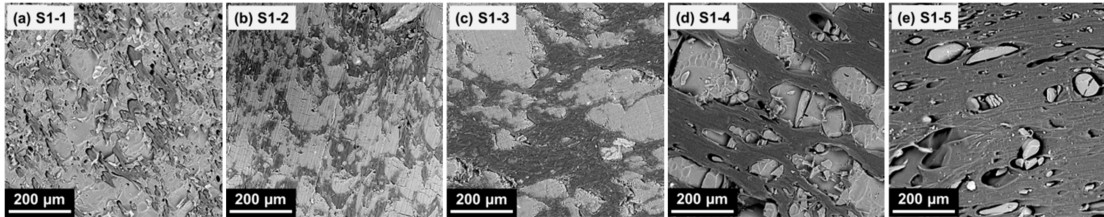

**Figure 1.** SEM micrographs of the cross sections of the injection moulded test specimens of the compounds produced in Series 1. Bright areas indicate novolac-based phase, dark areas indicate LDPE. LDPE content increases from left to right: 20 m%, 30 m%, 40 m%, 50 m%, 70 m%.

The phases in the SEM images were identified via EDX analysis; as an example, an image of the sample S1-5 (with a larger magnification than in Figure 1), with the marked spots for the EDX analysis and the results from the analysis, is shown in Figure 2. The elemental distribution at Pos. 1 roughly corresponds to the carbon–oxygen ratio found in phenol–formaldehyde resins: At a phenol–formaldehyde molar ratio of 1:1, the carbon–oxygen elemental ratio would be 87.5:12.5, but as novolacs contain an excess of phenol, the percentage of oxygen is slightly higher. In contrast, at Pos. 2, only carbon is detected, thereby clearly identifying the LDPE phase (N.B. Hydrogen atoms are not detected by EDX).

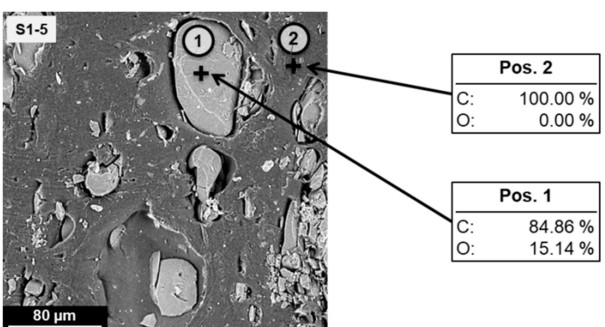

**Figure 2.** SEM micrograph of the cross section of an injection moulded test specimen of the formulation S1-5 with spots (1) and (2) indicating positions for the EDX analysis. The detected atomic percentages of carbon (C) and oxygen (O) are shown for each position.

The identification of the phases from the EDX analysis was verified by Raman mapping. Despite the strong fluorescence caused by the aromatic rings in the novolac, LDPE could be identified due to the C–H stretching signal between 2800 and 2900 $cm^{-1}$ [9]. The components could therefore be clearly distinguished (Figure 3).

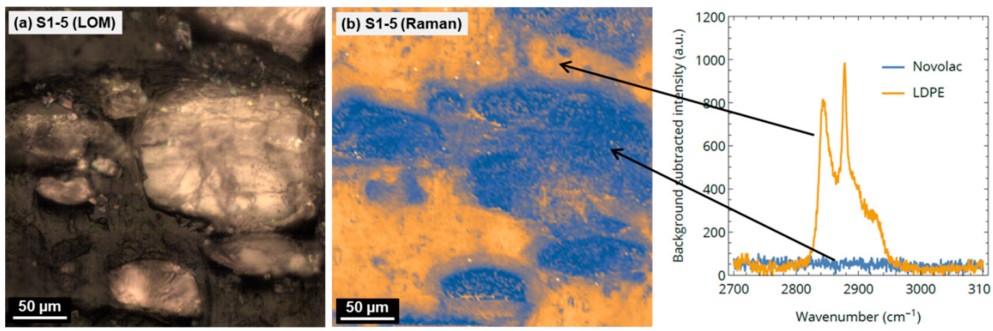

**Figure 3.** (**a**) Light optical microscopic (LOM) image of S1-5; (**b**) Raman mapping image of the same sample position, and background-corrected Raman spectra of LDPE and novolac. Blue regions in the Raman image indicate novolac (bright in LOM image), orange regions indicate LDPE (dark in LOM image).

### 3.1.2. Mechanical Properties

The results from the three-point bending test and the impact test are listed in Table 2. As expected, the impact strength increases with increasing LDPE content, whereas the opposite is true for the flexural modulus: while the LDPE can take up much more impact energy than the brittle novolac-based phase, a large amount of "soft" material obviously also reduces the stiffness of the compounds. The trends of the modulus and the impact strength in dependence of the LDPE content are also illustrated in Figure 4. This graph highlights not only the sharp decrease of the modulus from S1-1 to S1-2, which is likely connected to the phase inversion, but also the strong increase of the impact strength between S1-4 and S1-5, which is not only caused by the higher amount of LDPE itself, but also by the larger undisturbed bulk of the impact-resistant LDPE phase.

In contrast to the stiffness and the impact strength, there is no clear trend regarding the influence of the LDPE content on the flexural strength.

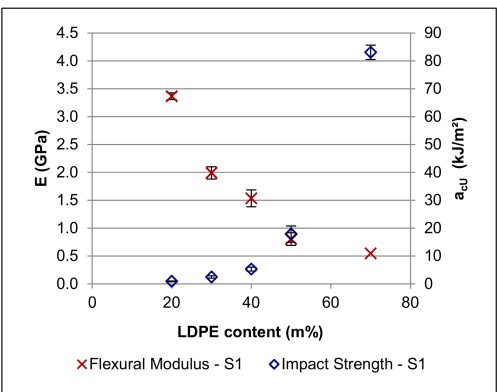

**Figure 4.** Modulus (*E*) and Charpy impact strength ($a_{cU}$) of the formulations of Series 1.

### 3.2. Series 2—Influence of Coupling Agent and Lubricant

The effect of the addition of processing additives, i.e., a coupling agent (Mowital) and a lubricant (Naftosafe), on the compound properties was investigated in Series 2. Based on the results from Series 1, the formulation S1-2, i.e., the one with a (novolac + HMTA)–LDPE ratio of 70:30, was selected as basic formulation for this series because the observed onset for a phase inversion was considered to provide the biggest potential for interesting morphological effects of the additives.

The compositions of the formulations prepared for Series 2 are shown in Table 3. It should be noted that, while the absolute percentages of (novolac + HMTA) and LDPE change depending on the concentration of the additives, their mass ratio remains at 70:30 for all samples. The series consisted of three compounds with different contents of the coupling agent Mowital B30T (2.5 m%, 5

m%, and 10 m%, respectively), two compounds with the lubricant Naftosafe PHX 369D (3 m% and 6 m%, respectively), and two compounds containing both additives (5 m% Mowital/3 m% Naftosafe, and 5 m% Mowital/6 m% Naftosafe, respectively).

**Table 3.** List of the formulations prepared for Series 2. Explanation of the nomenclature *S2-2_M.N*: S2-2 → Series 2, and (novolac + HMTA)–LDPE mass ratio = 70:30 as defined in Series 1; M.N → presence of Mowital (M) and Naftosafe (N), with higher numbers representing larger amounts.

| Sample Code | | Novolac | HMTA | LDPE | Mowital | Naftosafe |
|---|---|---|---|---|---|---|
| S2-2_1.0 | (m%) | 65.5 | 2.7 | 29.3 | 2.5 | - |
| | (vol%) | 59.2 | 2.3 | 35.9 | 2.6 | - |
| S2-2_2.0 | (m%) | 63.8 | 2.7 | 28.5 | 5.0 | - |
| | (vol%) | 57.6 | 2.3 | 35.0 | 5.1 | - |
| S2-2_3.0 | (m%) | 60.5 | 2.5 | 27.0 | 10.0 | - |
| | (vol%) | 54.5 | 2.1 | 33.1 | 10.2 | - |
| S2-2_0.1 | (m%) | 65.2 | 2.7 | 29.1 | - | 3.0 |
| | (vol%) | 58.6 | 2.3 | 35.6 | - | 3.5 |
| S2-2_0.2 | (m%) | 63.2 | 2.6 | 28.2 | - | 6.0 |
| | (vol%) | 56.5 | 2.2 | 34.3 | - | 7.0 |
| S2-2_2.1 | (m%) | 61.8 | 2.6 | 27.6 | 5.0 | 3.0 |
| | (vol%) | 55.5 | 2.2 | 33.7 | 5.1 | 3.5 |
| S2-2_2.2 | (m%) | 59.8 | 2.5 | 26.7 | 5.0 | 6.0 |
| | (vol%) | 53.5 | 2.1 | 32.4 | 5.1 | 6.9 |

In comparison with the basic formulation S1-2, the addition of Mowital without Naftosafe led to an increase of the recorded torque and melt pressure values and generally to a less stable compounding process. Interestingly, the highest values of torque, melt pressure, and melt temperature were recorded with a Mowital content of 2.5 m%, and decreased with increasing Mowital content. The same trend was observed for the maximum pressures during injection moulding (cf. Table 4).

Conversely, the addition of 3 m% Naftosafe led to a more stable compounding process with reduced values of torque, melt pressure, and melt temperature, as well as a lower injection pressure. This was the case regardless whether or not Mowital was present. A further increase of the Naftosafe content to 6 m% only had a little effect on the compounding process, but without Mowital, it unexpectedly resulted in a higher injection pressure.

**Table 4.** Recorded processing parameters for the compounds, and mechanical properties of the green bodies of Series 2. The properties of the basic formulation S1-2 are listed as a reference.

| Sample Code | *Compounding (145 °C)* | | | *Injection Moulding* | *Mechanical Properties* | | |
|---|---|---|---|---|---|---|---|
| | Torque | Melt Temp | Melt Pressure | Max. Injection Pressure | Flexural Modulus | Flexural Strength | Impact strength |
| | (Nm) | (°C) | (bar) | (bar) | (GPa) | (MPa) | (kJ/m$^2$) |
| *S1-2* | *18* | *158* | *7–30* | *970–1000* | *1.99 ± 0.11* | *8.8 ± 0.2* | *2.5 ± 0.4* |
| S2-2_1.0 | 25–28 | 164 | 48–67 | 1000 | 1.67 ± 0.04 | 9.3 ± 0.1 | 2.1 ± 0.5 |
| S2-2_2.0 | 20 | 160 | 50–65 | 920 | 1.63 ± 0.12 | 8.8 ± 0.5 | 2.6 ± 0.3 |
| S2-2_3.0 | 18–25 | 156 | 5–55 | 610 | 2.30 ± 0.13 | 12.3 ± 1.1 | 1.1 ± 0.1 |
| S2-2_0.1 | 10 | 153 | 5–15 | 490 | 2.68 ± 0.11 | 13.4 ± 1.4 | 1.4 ± 0.2 |
| S2-2_0.2 | 10 | 153 | 5–15 | 610 | 1.77 ± 0.09 | 16.3 ± 0.6 | 2.2 ± 0.2 |
| S2-2_2.1 | 16–18 | 156 | 9–43 | 570 | 2.56 ± 0.09 | 15.3 ± 1.1 | 1.3 ± 0.2 |
| S2-2_2.2 | 10–20 | 156 | 10–25 | 520 | 2.71 ± 0.03 | 16.4 ± 1.7 | 1.7 ± 0.2 |

### 3.2.1. Morphology

The SEM images of the Series 2 compounds are shown in Figure 5. The addition of low amounts of Mowital (2.5 m% and 5 m% → S2-2_1.0 and S2-2_2.0) resulted in a more clearly defined phase separation

compared to the basic formulation S1-2, with the LDPE forming the matrix. The dispersion of the novolac is somewhat coarser with 5 m% Mowital content than with 2.5 m%, and becomes finer again at 10 m% (S2-2_3.0); in the latter sample, the phase separation is also less well-defined. In combination with the observed effect of different Mowital concentrations on the processing parameters, as discussed above, these findings suggest that the addition of Mowital has two competing effects: on the one hand, the improved phase compatibility resulting from the presence of the coupling agent favours a finer dispersion and should, at least in the absence of a hardening agent, facilitate processing; and on the other hand, the increased phase adhesion increases the friction during compounding, which in turn results in a higher melt temperature and consequently an accelerated hardening of the novolac-based phase, which reduces the phase compatibility again.

The addition of Naftosafe also has a strong influence on the morphology. Without Mowital, adding 3 m% Naftosafe (S2-2_0.1) leads to a significantly finer novolac–LDPE distribution compared to the basic compound S1-2, and also to less clearly defined continuous and dispersed phases. This is a consequence of the lower melt temperature and thus of a reduced hardening of the novolac-based phase. However, if the Naftosafe content is further increased to 6 m% (S2-2_0.2), the morphology changes again and a distinct LDPE matrix with a dispersed novolac-based phase is formed. In this case, the phase distribution is coarser than in S2-2_0.1, but much finer than in the basic formulation S1-2. This change is also reflected in the abovementioned increase in injection pressure. This suggests that, while leading to comparatively gentle processing conditions, a high Naftosafe concentration tends to reduce the phase compatibility of novolac and LDPE.

In the presence of 5 m% Mowital, the addition of Naftosafe also leads to a much finer phase distribution and less clearly defined continuous and dispersed phases. The morphology of the sample S2-2_2.2 (5 m% Mowital and 6 m% Naftosafe) actually looks very similar to the sample S2-2_0.1. It is obvious that the presence of the lubricant mitigates the effect of the coupling agent which promotes the hardening of the novolac-based phase. From these observations, it follows that a fine balance of the amounts of coupling agent and lubricant is necessary in order to achieve a good phase compatibility of the novolac–LDPE-based compounds.

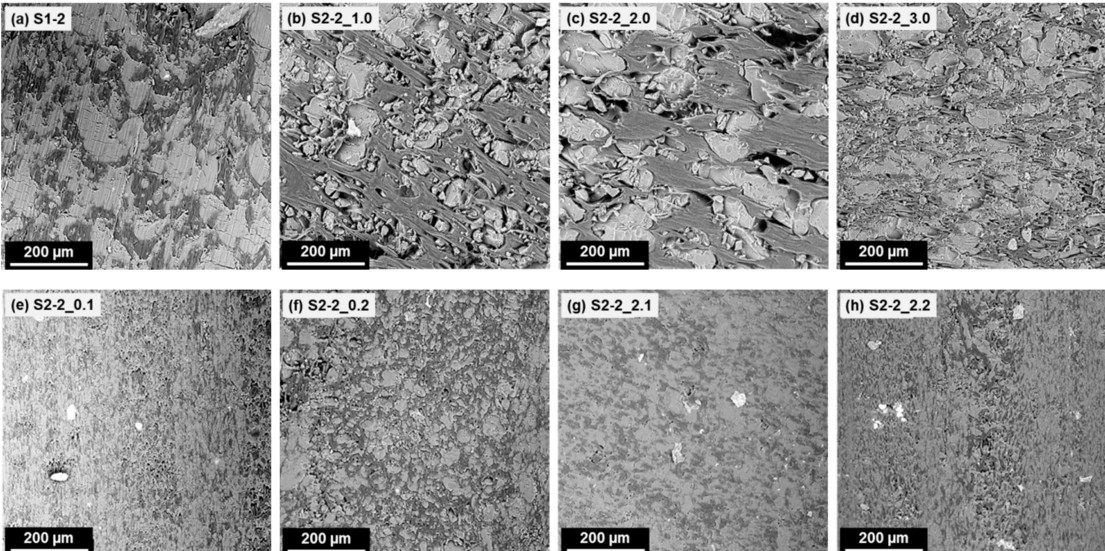

**Figure 5.** SEM images of the compounds with a 70:30 (novolac + HMTA)–LDPE ratio and different amounts of additives. Bright areas—novolac-based phase; dark areas—LDPE. Mowital and Naftosafe cannot be distinctly identified in the SEM images. **Top row:** samples containing 0 m%/2.5 m%/5 m%/10 m% Mowital and no Naftosafe; **Bottom row:** samples containing no Mowital and 3 m%/6 m% Naftosafe, and samples containing 5 m% Mowital and 3 m%/6 m% Naftosafe.

3.2.2. Mechanical Properties

The results from the mechanical characterisation are summarised in Table 4. In the absence of Naftosafe, no clear effect of Mowital on the flexural strength and the impact strength is observed for contents of up to 5 m%, whereas the flexural modulus is slightly lower than in the reference formulation S1-2. At 10 m% Mowital content, the impact strength drops sharply and the flexural modulus and flexural strength increase; these effects correlate to the change in the morphology as shown above.

The addition of 3 m% of Naftosafe significantly reduces the impact strength and increases the flexural modulus and strength, regardless of the presence of Mowital. Consequently, both formulations S2-2_0.1 and S2-2_2.1 show very similar flexural moduli and impact strengths. The same is true for the formulation S2-2_2.2 containing 6m% Naftosafe, which is an expected result because of the very similar morphology to S2-2_0.1. However, the flexural strength of S2-2_2.1 and S2-2_2.2 is slightly higher than that of S2-2_0.1, which indicates a stronger phase adhesion because of the presence of the Mowital.

Compared to the other three formulations containing Naftosafe, the formulation S2-2_0.2 (0 m% Mowital and 6 m% Naftosafe) exhibits a much lower flexural modulus and a much higher impact strength, which can be attributed to the changes of the morphology.

*3.3. Series 3—Addition of Fibres*

Out of the samples characterised in Series 2, S2-2_0.1 and S2-2_2.2 were chosen as basic compositions for the addition of fibres due to their similar morphology and their comparatively fine novolac–LDPE dispersion. For comparison, the formulation S1-2, which had a very coarse morphology, was also chosen. The corresponding fibre-reinforced formulations were named S3-2, S3-2_0.1, and S3-2_2.2, respectively. The exact compositions of these samples are shown in Table 5. It is important to note that for each formulation, despite the addition of a total of 30 m% fibres in all cases, the ratios of (novolac + HMTA), LDPE, and (where applicable) the additives remained unchanged compared to their respective basic formulations S1-2, S2-2_0.1, and S2-2_2.2.

**Table 5.** List of the formulations prepared for Series 3. Explanation of the nomenclature *S3-2_M.N*: S3-2 → Series 3, and (novolac + HMTA)–LDPE mass ratio = 70:30 as defined in Series 1; M.N → presence of Mowital (M) and Naftosafe (N), with higher numbers representing larger amounts.

| Sample Code | | Novolac | HMTA | LDPE | Mowital | Naftosafe | Cellulose | Wood |
|---|---|---|---|---|---|---|---|---|
| S3-2 | (m%) | 47.0 | 2.0 | 21.0 | - | - | 15.0 | 15.0 |
| | (vol%) | 45.9 | 1.8 | 27.8 | - | - | 12.2 | 12.2 |
| S3-2_0.1 | (m%) | 45.6 | 1.9 | 20.4 | - | 2.1 | 15.0 | 15.0 |
| | (vol%) | 44.4 | 1.7 | 26.9 | - | 2.6 | 12.2 | 12.2 |
| S3-2_2.2 | (m%) | 41.9 | 1.7 | 18.7 | 3.5 | 4.2 | 15.0 | 15.0 |
| | (vol%) | 40.5 | 1.6 | 24.6 | 3.8 | 5.3 | 12.1 | 12.1 |

Because of the much higher friction during compounding due to the fibres, a lower temperature profile (i.e., a barrel temperature of 135 °C) had to be used, whereas the other processing parameters remained unchanged. The recorded processing parameters are summarised in Table 6. By far the lowest melt temperature, melt pressure, and injection pressure were, surprisingly, recorded for the formulation S3-2, i.e., without Mowital and Naftosafe. While clearly the highest melt temperature and melt pressure were recorded for S3-2_2.2, the injection pressure for this formulation was slightly lower than for S3-2_0.1; it should however be noted that the difference in injection pressure between these two samples cannot be considered significant.

**Table 6.** Recorded processing parameters for the compounds, and mechanical properties of the green bodies of Series 3.

| Sample Code | *Compounding (135 °C)* | | | *Injection Moulding* | *Mechanical Properties* | | |
|---|---|---|---|---|---|---|---|
| | Torque | Melt Temp | Melt Pressure | Max. Injection Pressure | Flexural Modulus | Flexural Strength | Impact Strength |
| | (Nm) | (°C) | (bar) | (bar) | (GPa) | (MPa) | (kJ/m$^2$) |
| S3-2 | 21 | 150 | 16 | 670 | 5.33 ± 0.10 | 34.8 ± 0.3 | 3.2 ± 0.1 |
| S3-2_0.1 | 20–30 | 153–163 | 15–60 | 1450 | 3.36 ± 0.03 | 20.9 ± 0.6 | 3.7 ± 0.3 |
| S3-2_2.2 | 25–30 | 161–164 | 30–110 | 1300 | 3.71 ± 0.04 | 27.0 ± 0.9 | 4.4 ± 0.3 |

### 3.3.1. Morphology

As shown in Figure 6, the addition of fibres leads to complicated morphologies without a clearly defined continuous phase. While the wood particles are clearly visible in the SEM images due to their size and shape, it is difficult to identify the cellulose fibres due to their much smaller diameter. Generally, the fillers appear slightly brighter in the images than the novolac because of their higher density (approximately 1.5 g/cm$^3$ vs. 1.25 g/cm$^3$). Regarding the distribution of the polymeric phases and the fibres, there seems to be little difference between the formulations S3-2_0.1 and S3-2_2.2; a similar observation has already been made for the corresponding samples without fibres. In contrast, the morphology of S3-2 is much coarser, and the wood and cellulose particles appear to be embedded only in the novolac-based phase, with a clearer separation to the LDPE regions than in the other two formulations.

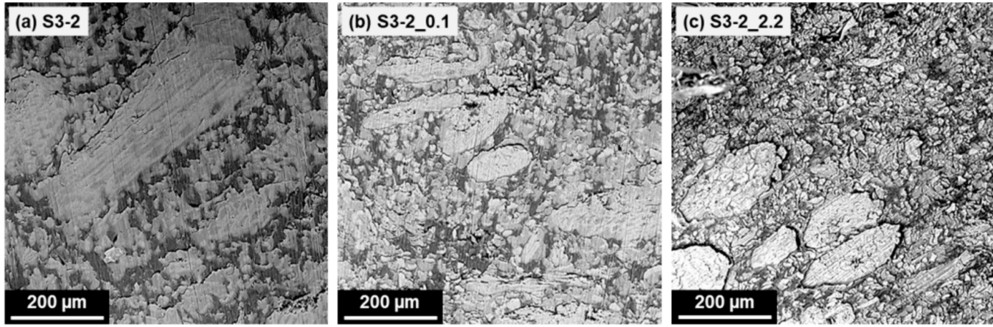

**Figure 6.** SEM images of the formulations prepared in Series 3. Brightest areas—wood particles (large particles, approximately 200–500 μm in length) and cellulose fibres (small, roughly circular to elliptical shapes, approximately 15 μm in diameter); light grey areas (slightly darker than fillers)—novolac-based phase; dark areas—LDPE. Mowital and Naftosafe cannot be distinctly identified in the SEM images.

### 3.3.2. Mechanical Properties

The mechanical properties of the materials of Series 3 are summarised in Table 6. A comparison of the flexural moduli and impact strengths of the new formulations S3-2, S3-2_0.1, and S3-2_2.2 with the respective values of their fibre-free counterparts S1-2, S2-2_0.1, and S2-2_2.2 is illustrated in Figure 7. As expected, the addition of the fibres leads to a significant increase of the impact strength and the flexural modulus. Surprisingly, the modulus of S3-2 is much higher and its impact strength is lower than the respective values of S3-2_0.1 and S3-2_2.2, despite the corresponding fibre-free sample S1-2 having the lowest modulus and the highest impact strength of the materials under consideration. The recorded values of the flexural strength follow the same trend as the flexural modulus (cf. Table 6). S3-2_2.2 shows better overall mechanical properties than S3-2_0.1, which can be attributed to the better adhesion between the phases due to the presence of Mowital.

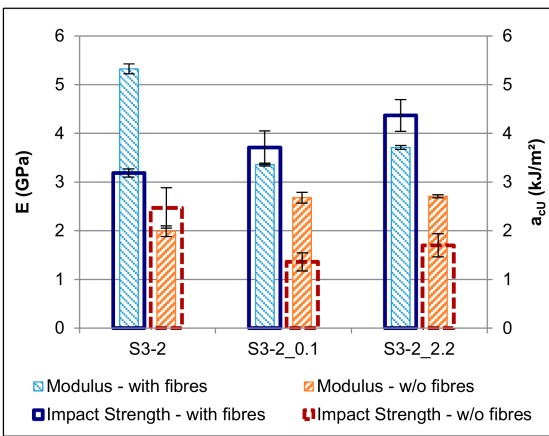

**Figure 7.** Effect of the addition of fibres on the flexural modulus (*E*) and on the impact strength ($a_{cU}$).

### 3.4. Carbonisation and Liquid Silicon Infiltration

From the green bodies of the compounds prepared in Series 3, porous carbon specimens were produced by pyrolysis, during which the specimens underwent significant shrinkage in all directions; in the following discussion, the carbon samples are identified by the suffix "_C". From these, SiC ceramic specimens (suffix "_SiC" in the following discussion) were obtained via liquid silicon infiltration. Figure 8 shows an exemplary comparison of a green body, a porous carbon template, and a ceramic specimen.

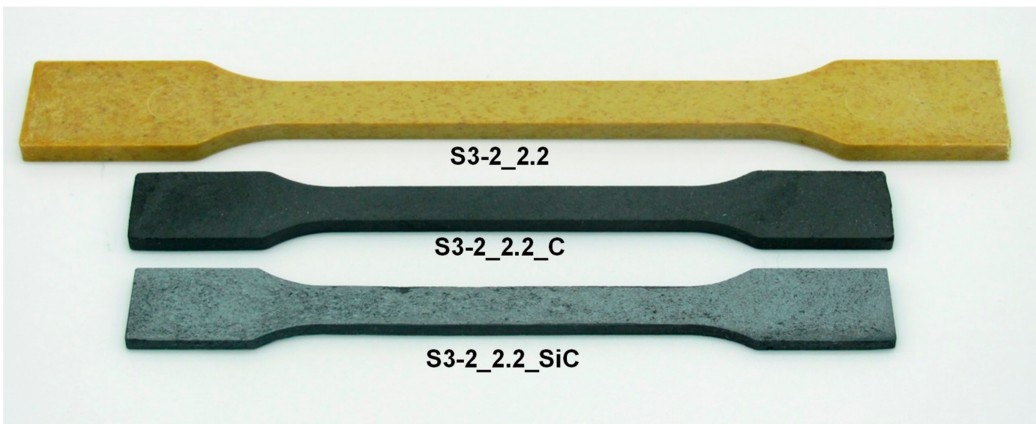

**Figure 8.** Comparison of a green body (top), a porous carbon template (middle), and a C/Si/SiC ceramic specimen obtained from the formulation S3-2_2.2.

### 3.4.1. Morphology

The morphologies of the carbon and the ceramic specimens were analysed using optical light microscopy. In Figure 9, the SEM images of the corresponding green bodies with and without fibres, and the LOM images of the carbon and ceramic specimens are compared. From the LOM images, the volumetric fractions of the phases (carbon and pores in the carbon specimens; SiC, silicon, carbon, and pores in the ceramic specimens) were calculated. The results, as well as the bulk densities of the carbon and the ceramic specimens, are summarised in Table 7.

As in the corresponding green bodies, no significant differences were observed in the morphologies of the samples S3-2_0.1_C and S3-2_2.2_C, except for a slightly higher presence of small, elongated voids in the case of S3-2_0.1_C. These voids are likely caused by a lower adhesion of the components in the green bodies due to the absence of a coupling agent. The otherwise similar morphologies were also reflected in the respective total pore volume and in the equal raw densities of the carbon specimens. In contrast, the carbon specimens S3-2_C showed a much coarser pore distribution and a higher total

pore volume, and correspondingly a much lower density. In all three samples, the remnants of the wood particles are clearly visible.

From the microscopic images, it becomes clear that the areas of pure silicon in the ceramic specimens correspond to the voids present in the carbon specimens, and the areas of residual carbon correspond mostly to the remnants of the wood particles. A comparison of S3-2_0.1_SiC and S3-2_2.2_SiC shows that despite the very similar volume fractions of the different phases, their distribution throughout the specimens is somewhat different, especially in the case of the pure silicon: the elongated voids present in S3-2_0.1_C led to the presence of larger domains of silicon in S3-2_0.1_SiC, whereas in S3-2_2.2_SiC the silicon is more finely distributed.

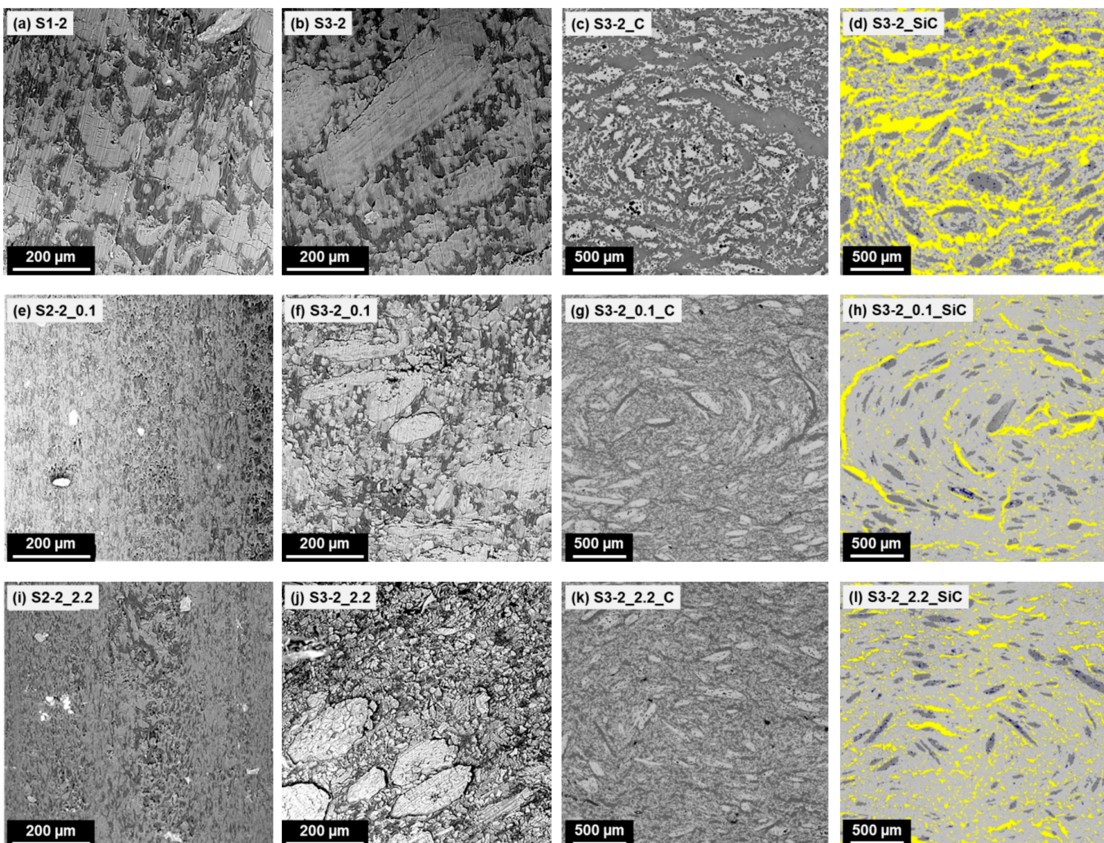

**Figure 9.** Comparison of green bodies without fibres (left; SEM), green bodies with fibres (second from left; SEM), porous carbon specimens (third from left; LOM), and silicon carbide specimens (right; LOM, false-colour display). **Left column**: bright areas—novolac; dark areas—LDPE. **2nd column**: brightest areas—wood particles (large particles, approximately 200–500 μm in length) and cellulose fibres (small, roughly circular to elliptical shapes, approximately 15 μm in diameter); light grey areas (slightly darker than fillers)—novolac; dark areas—LDPE. Mowital and Naftosafe cannot be distinctly identified in the SEM images. **3rd column**: bright areas—carbon; dark areas—pores and/or voids. **Right column**: light grey areas—silicon carbide; yellow areas—silicon; dark grey areas—carbon; dark blue spots—pores.

**Table 7.** Raw densities, phase distributions, and mechanical properties of the C and C/Si/SiC specimens.

| Sample Code | Raw Density | *Phase Distribution* | | | | *Mechanical Properties* | |
|---|---|---|---|---|---|---|---|
| | | Pores | Carbon | Si | SiC | Flexural Modulus | Flexural Strength |
| | (g/cm³) | (vol%) | (vol%) | (vol%) | (vol%) | (GPa) | (MPa) |
| S3-2_C | 0.64 | 63.9 | 26.1 | - | - | 6.1 ± 0.1 | 19.3 ± 0.3 |
| S3-2_0.1_C | 0.71 | 52.5 | 47.5 | - | - | 2.9 ± 0.3 | 8.4 ± 0.9 |
| S3-2_2.2_C | 0.71 | 53.1 | 46.9 | - | - | 5.5 ± 0.3 | 16.7 ± 1.0 |
| S3-2_SiC | 2.43 | 0.5 | 27.4 | 28.9 | 43.2 | 144.1 ± 35.5 | 118.3 ± 22.7 |
| S3-2_0.1_SiC | 2.71 | 0.3 | 15.1 | 8.8 | 75.8 | 225.8 ± 3.9 | 168.0 ± 28.7 |
| S3-2_2.2_SiC | 2.71 | 0.4 | 16.4 | 10.7 | 72.5 | 230.6 ± 4.2 | 190.8 ± 34.7 |

### 3.4.2. Mechanical Properties

The flexural properties of the carbon specimens and the C/Si/SiC ceramics are shown in Table 7 and illustrated in Figure 10. Despite the similar pore structure of the carbon specimens S3-2_0.1_C and S3-2_2.2_C, the former had a much lower stiffness and strength than the latter, whereas the best overall flexural properties were shown by S3-2_C. While this is surprising, it reflects the properties already shown by the green bodies. It appears that the presence of a lubricant significantly deteriorates the flexural properties, but that this is at least partly counteracted by the presence of a coupling agent, and that these effects are retained even after pyrolysis.

As expected, the very coarse morphology and the relatively small fraction of actual SiC in the specimens led to comparatively poor stiffness and strength values of S3-2_SiC. In contrast, S3-2_0.1_SiC and S3-2_2.2_SiC showed significantly better flexural properties. However, while their respective moduli were similar to each other, S3-2_2.2_SiC exhibited a much higher flexural strength. This may be explained by the finer distribution of the pure Si in the ceramic specimens, as opposed to the larger Si domains in the case of S3-2_0.1_SiC.

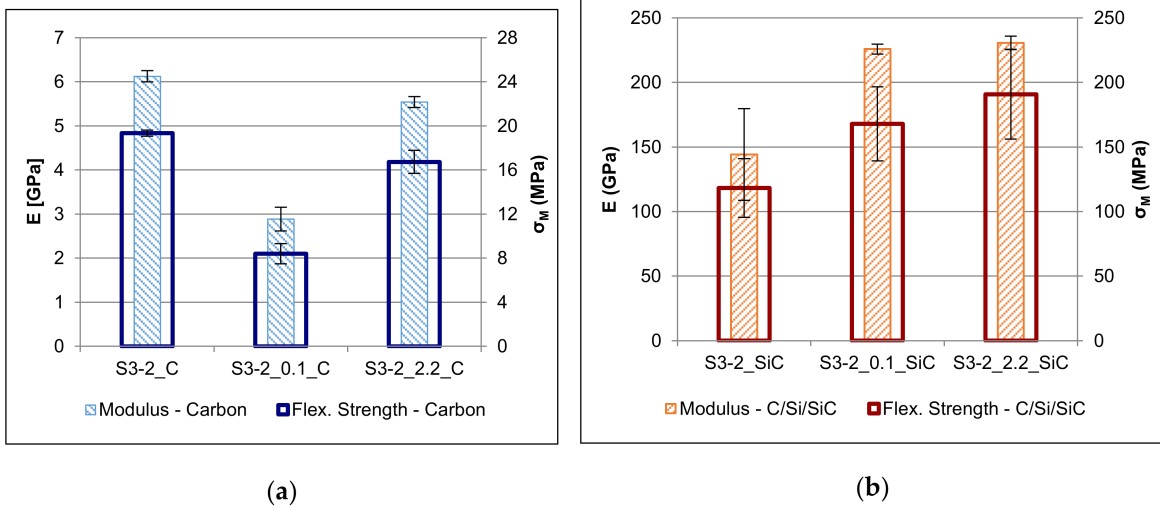

(**a**) (**b**)

**Figure 10.** Flexural modulus (*E*) and strength ($\sigma_M$) of the porous carbon specimens (**a**) and of the C/Si/SiC ceramics (**b**).

## 4. Summary and Conclusions

The influence of the composition of novolac–LDPE-based compounds on the morphology and mechanical properties of the green bodies, and on the properties of the subsequently produced carbon specimens and C/Si/SiC ceramics, was investigated.

Without further additives, the novolac–LDPE-based mixtures generally showed very bad phase compatibility which resulted in coarse morphologies. However, by adding a suitable combination of a coupling agent and a lubricant, the phase compatibility could be significantly improved, resulting in a much finer distribution.

The LDPE content and the morphology strongly influenced the mechanical properties of the compounds without fibres. A higher LDPE content led to an increase of the impact strength and a reduction of the stiffness. If LDPE formed the continuous phase, a coarse phase dispersion resulted in a higher impact strength because of the larger bulk of undisturbed impact resistant material. No obvious influence of the phase dispersion on the stiffness was observed. The addition of fibres resulted in a significant improvement of the mechanical properties, which were again strongly influenced by the phase distribution and the adhesion between the phases.

The porous carbon specimens with the coarsest pore structure, obtained from the formulation without additives, surprisingly showed the best flexural properties. However, it was shown that a fine pore distribution in the carbon templates is important in order to achieve a high conversion of the carbon into silicon carbide, because the presence of large pores or voids results in a high amount of pure silicon being embedded into the structure, which in turn is detrimental to the stiffness and strength of the ceramics specimens.

In summary, it may be concluded that in order to achieve good mechanical properties of the bio-based silicon carbide ceramics, one must ensure a good phase compatibility of the components in the green bodies. This, in turn, can be achieved by a careful selection of the composition of the polymeric matrix, as well as a proper setting of the processing parameters. One of the most important factors influencing the homogeneity of the green bodies is the hardening that the novolac-based phase undergoes during processing. Therefore, a systematic investigation of the degree of curing depending on the composition and processing conditions would be a very interesting topic for a future study.

**Author Contributions:** Conceptualisation, M.M. and C.F.; investigation, M.M., F.S.d.S., I.B., and A.H.; methodology, M.M., I.B., and C.F.; project administration, C.F.; resources, D.S.; supervision, M.M. and C.F.; visualisation, M.M. and A.H.; writing—original draft, M.M.; writing—review and editing, M.M., C.F., and D.S.

**Funding:** This work was performed within the project Biocarb-K, supported by the European Regional Development Fund (EFRE) and the province of Upper Austria through the programme IWB 2014-2020.

**Conflicts of Interest:** The authors declare no conflict of interest. The funders had no role in the design of the study; in the collection, analyses, or interpretation of data; in the writing of the manuscript; or in the decision to publish the results.

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
