# Peer review of "Morphology and Characterisation of Novolac–LDPE-Based Mixtures as Matrix for Injection Moulded Green Bodies for Bio-Based SiC Ceramics"

_ceramics, doi:10.3390/ceramics2030041_

Round 1

Reviewer 1 Report

It is interesting work, but in many places it is chaotic and incomplete so it should be corrected,

•              “The identification of the phases done via EDX was verified by Raman mapping.” – where are the EDX results?

•             Please ad in methodological section the numbers of samples for mechanical tests.

•             Fig 3. – I suggest to use another type of the plot – here we can’t see e.g. SD bars. What is on “x” axis? Maybe tables will be better?

•             You have shown impact strength and flexural modulus. So if you realized three-point bending test, test you can also show flexural strength. Please do it. Subsection 3.3 also please add flexural strength results.

•             “The results of the mechanical characterisation are summarised in ...”  (line 205-211)- we don’t have this reference. Those results were published before? If yes, this part should be removed from results section, but rater can be summarized in discussion and methodological section. The same in subsection 3.3. You should divide previously published and commented result and new achievements. At this moment it is hard to say what is new in this work.

•             Table 3. – where are standard deviations values? Here you have shown flexural strength and modulus, but you omitted impact strengths. The results are the same like in Fig. 8? It yes, please removed one of them.

•             There is no discussion in this work, which significantly reduces the scientific value of work, which becomes a report from the conducted research.

•             What are expected properties for materials what was considered (discussion)?

•             Please prepare statistical analysis of the results

Other

•             Figure 7 should not be before Fig. 5 and 6.

•             Please add a short subsection about potential application of your experimental materials.

•             Some references are lost: line 82, 105,137,206, 217, 220, 258,287.

Reviewer 2 Report

See comments in attached file.

Round 2

Reviewer 1 Report

The article has been improved in the basic scope and can be accepted, but I still think that conducting statistical analysis would improve the quality of work. It has nothing to do with optimization, as the authors suggest.However, in the case of the presented results, I do not treat it obligatorily, so if the authors made such a decision to the detriment of their work, I accept it.

Author Response

We do not think a statistical analysis of the presented results would be feasible. As the reviewer is ready to accept the manuscript without it, we prefer to leave it as it is.

Reviewer 2 Report

Changes to the manuscript are sufficient to support acceptance with minor corrections. Paper should be edited for moderate changes to grammar - many sentences run-on and/or seem to contain sentence fragments.

Author Response

We have slightly rephrased some passages in order to eliminate the run-on sentences as far as possible. We have also made minor changes to the wording in some cases and correcred a few mistakes.